# Regulation of Satellite Cells Functions during Skeletal Muscle Regeneration: A Critical Step in Physiological and Pathological Conditions

**DOI:** 10.3390/ijms25010512

**Published:** 2023-12-29

**Authors:** Giorgia Careccia, Laura Mangiavini, Federica Cirillo

**Affiliations:** 1Department of Biosciences, University of Milan, 20133 Milan, Italy; giorgia.careccia@unimi.it; 2IRCCS Istituto Ortopedico Galeazzi, 20161 Milan, Italy; laura.mangiavini@unimi.it; 3Department of Biomedical Sciences for Health, University of Milan, 20133 Milan, Italy; 4IRCCS Policlinico San Donato, 20097 San Donato Milanese, Italy; 5Institute for Molecular and Translational Cardiology (IMTC), 20097 San Donato Milanese, Italy

**Keywords:** satellite cells, skeletal muscle regeneration, sarcopenia, muscular dystrophy, cancer cachexia

## Abstract

Skeletal muscle regeneration is a complex process involving the generation of new myofibers after trauma, competitive physical activity, or disease. In this context, adult skeletal muscle stem cells, also known as satellite cells (SCs), play a crucial role in regulating muscle tissue homeostasis and activating regeneration. Alterations in their number or function have been associated with various pathological conditions. The main factors involved in the dysregulation of SCs’ activity are inflammation, oxidative stress, and fibrosis. This review critically summarizes the current knowledge on the role of SCs in skeletal muscle regeneration. It examines the changes in the activity of SCs in three of the most common and severe muscle disorders: sarcopenia, muscular dystrophy, and cancer cachexia. Understanding the molecular mechanisms involved in their dysregulations is essential for improving current treatments, such as exercise, and developing personalized approaches to reactivate SCs.

## 1. Introduction

Skeletal muscle is composed of syncytial muscle fibers, each characterized by hundreds of myonuclei that support efficient contractility by releasing mRNA and promoting protein synthesis [1,2]. Skeletal muscle requires large amounts of energy and proteins for its functions but is also a source of energy during starvation and disease and, therefore, shows high plasticity. Cell size generally results from the balance between catabolism and anabolism, which are equally represented in homeostasis.

On the other hand, the well-being of skeletal muscle also depends on the ability to repair and regenerate new muscle fibers after traumatic events, thanks to the presence of adult stem cells, the satellite cells (SCs), first identified by Alexander Mauro [3]. SCs have two essential functions: (i) the ability to self-renew, which is essential to maintain and replenish the stem cell pool, and (ii) the capacity to differentiate into myogenic progenitor cells [4]. Defects in self-renewal ability result in a depleted SCs pool, impairing muscle regeneration capacity.

After injury, SCs undergo asymmetric division necessary to preserve the SCs pool, maintaining an undifferentiated mother cell that returns to a quiescent state and providing committed progenies [5]. During this phase, asymmetric co-segregation of immortal and younger DNA strands occurs in different daughter cells [6]. Specifically, the immortal DNA strand is co-segregated in daughter cells expressing the stem cell marker Sca1. In contrast, the younger strand is allocated to differentiated daughter cells expressing the differentiation marker Desmin [6]. The committed SCs become myoblasts, which further proliferate and eventually differentiate into postmitotic myocytes that fuse with existing fibers or each other to form new myofibers [7].

Nevertheless, some physiological and pathological conditions, such as aging, muscular dystrophy, and cancer cachexia, affect the ability of SCs to restore skeletal muscle tissue after damage, resulting in impaired regenerative machinery [8,9,10]. The molecular and biochemical mechanisms activated in these diseases are diverse and influence the number and functionality of SCs. In particular, age-related sarcopenia is associated with a decrease in SCs during aging, especially those associated with fibers II [11].

On the other hand, Duchenne muscular dystrophy is characterized by a progressive decline in functional SCs and their myogenic capacity, leading to the loss of myofibers and their replacement by fibrotic tissue [12,13]. Last but not least, cancer cachexia is a severe condition characterized by progressive body wasting and a remarkable weight loss of skeletal muscle [14].

In this review, we will discuss the role of SCs in controlling the skeletal muscle regeneration process, focusing on their functional and regulatory mechanisms, dysfunction, and therapeutic implications.

## 2. Biology of Satellite Cell in Regulating Muscle Homeostasis and Regeneration

The ability of skeletal muscle tissue to maintain its uniform architecture and functions is due to SCs, which reside in a defined anatomical hypoxic niche [15]. Upon injury, the oxygen concentration in muscle tissue and niches changes, and so does the expression of hypoxia-inducible factors (HIFs) [16,17]. In particular, HIF1α levels rapidly increase after cardiotoxin (CTX)-induced muscle injury, while HIF2α levels drop sharply after injury and gradually return to the baseline level after muscle regeneration [16]. Moreover, pharmacological activation of the HIF1α improves skeletal muscle regeneration and leads to the formation of bigger fibers [18,19].

In hypoxic niches, quiescent SCs express the paired-box transcription factors PAX7 and PAX3; the latter is particularly abundant in the diaphragm but much less in hindlimb muscles [20,21,22]. The differential expression of PAX3 in the tissues is probably due to the assumption that it plays a minor role in myogenesis in hindlimb muscles. On the other hand, PAX7 has been described as a master regulator of SCs’ function because it controls the expression of genes involved in SCs’ survival and proliferation and inhibits genes that trigger differentiation [23,24]. In addition to its role as a regulator, its activity is also modulated by post-translational modifications such as acetylation [25]. Although the expression of PAX7 is crucial to identify SCs in skeletal muscle tissue, it is now accepted that SCs are a heterogeneous population of stem cells. Indeed, in vivo experiments have demonstrated that SCs are characterized by the expression of PAX7 at different levels, which reflect multiple subpopulations of SCs with other functions. In particular, SCs characterized by a high expression of PAX7 (PAX7^High^) require a longer time to undergo first mitosis compared to SCs with a low expression of PAX7 (PAX7^Low^), which is also the reason why some SCs lose their ability to return to a quiescent state and become more committed to differentiation [26].

Moreover, SCs segregate their DNA asymmetrically, so the daughter cells that receive the template strand also maintain the expression of stem cell markers [27,28]. In addition, recent articles have identified different populations of SCs based on the expression of Gli factors. Gli1-positive SCs are more prone to proliferate and differentiate and contribute to the establishment of G^Alert^ [29]. Gli2 is expressed during muscle stem cell activation, while Gli3 plays a crucial role in establishing quiescence of muscle stem cells [30,31]. Therefore, the stemness of SCs is characterized by a complex mechanism controlled by multiple factors.

Myogenic progenitor cells, often called myoblasts, are highly proliferative cells characterized by the expression of myogenic regulatory factors (MRFs), especially MYOD, which is crucial for adult muscle regeneration [32,33]. Indeed, MYOD knockout (KO) mice show a delay in the onset of myogenic differentiation and an increased propensity for self-renewal, suggesting its role in the balance between self-renewal and differentiation during regeneration [34,35,36,37]. In this context, Fujita, R. and colleagues have recently published that SCs with a low expression level of MYOD are undifferentiated. In contrast, SCs with a high expression of MYOD showed commitment and differentiation signatures [38]. Moreover, several studies have shown that the over-expression of MYOD can reprogram the fate of many non-muscle cell types, such as fibroblasts, to a myogenic lineage when specifically over-expressed [39,40,41]. MYOD has been reported to regulate CDC6 expression early after activation and induce SCs to re-enter the cell cycle [42]. Depending on the level and activity of MYOD, activated SCs have two different fates: they can reduce MYOD level and self-renew, guaranteeing the maintenance of a pool of quiescent PAX7^+^ SCs; alternatively, SCs can maintain MYOD expression, leading to the downregulation of PAX7 and determining the commitment to differentiation [38,43]. Therefore, the tight balance between PAX7 and MYOD determines the fate of SCs.

Once SCs begin to differentiate into myoblasts, other specific differentiation markers are upregulated, such as MYOGENIN, different isoforms of myosin heavy chain (MHC), slow-twitch skeletal muscle troponin T (TNNT1), cardiac and slow-twitch skeletal muscle Ca^2+^-ATPase (ATP2A2), insulin-like growth factor-2 (IGF-2), fibroblast growth factor receptor 4 (FGFR4), nicotinic cholinergic receptor alpha polypeptide 1 (CHRNA1), and cardiac/slow-twitch skeletal muscle troponin C (TNCC) [44,45]. Noteworthy, MYOGENIN is a direct target of MYOD and promotes terminal differentiation of myogenic progenitor cells, which is associated with the downregulation of MYOD [20,32,46]. The involvement of several factors in the differentiation phase suggests that skeletal muscle regeneration is controlled by a complex mechanism in which each element plays a crucial role.

The final phase of this process requires the elongation of myocytes into multinucleated myotubes, their maturation into contractile myofibers, and the expression of adult MHC isoforms with temporally and spatially regulated expression patterns. In particular, skeletal muscles of different mammalian species contain three major myosin heavy-chain (MHC) isoforms: one “slow”, I-MHC, and three “fast” IIa-, IIb-, and IIx-MHCs [47]. The differential distribution of MHCs defines four major fiber types containing a single MHC isoform and several intermediate hybrid fiber populations containing both slow- and IIa-MHC, IIa- and IIx-MHC, or IIx- and IIb-MHC, characterized by different metabolic features. For example, the slow-oxidative unit mainly expresses a slow MHC gene known as slow type I. The fast-oxidative unit expresses a combination of the fast type IIa and IIx MHC genes. In contrast, the fast-glycolytic unit combines the fast IIb and IIx MHC genes [48]. The amount of MHC and its isoforms was associated with the type of innervation. Indeed, following denervation injury, slow fibers show a decrease in the content of I-MHC while the amount of IIa-MHC increases [49]. These changes are reversed 10 days after the reinnervation but with some significant differences depending on the fiber type [49]. In fast fibers, denervation leads to an enhancement in IIa-MHC and a reduction in IIb-MHC. In this case, reinnervation restores the ratio between the two fiber types as the IIa isoform continues to increase [49].

Several characters are involved in the proliferation and differentiation of SCs, including (i) fibro-adipogenic progenitor cells (FAPs), which have been defined as multipotent progenitor cells able to differentiate into fibroblasts, adipocytes, and possibly osteoblasts and chondrocytes, but not myoblasts, and (ii) inflammatory and immune cells, that secrete cytokines and factors determining the timing of regeneration [50]. In the quiescent state, FAPs are often localized near blood vessels outside the capillary basement membrane. In contrast, during muscle injury, FAPs proliferate and expand, creating a transient favorable environment that promotes SCs regeneration [50]. The deposition and secretion of type I and III collagen, fibronectin, elastin, proteoglycans, and laminin are essential for maintaining the overall integrity of the tissue and providing a scaffold for newly formed fibers [51]. Expansion of FAPs is critical during regeneration to sustain SCs differentiation in a paracrine manner and to maintain the SCs pool [52]. Among the molecules secreted by FAPs, inducible signaling pathway protein 1 (WISP1) of Wnt family 1 (WNT1) plays an essential role in regulating SCs expansion and asymmetric commitment to myogenic differentiation [53].

On the other hand, almost all phases of muscle regeneration are tightly regulated by acute inflammation and immune cells, and changes in the inflammatory response modify the course of muscle regeneration. Indeed, inhibition of inflammation destroys muscle repair. In contrast, a stronger initial inflammatory response results in accelerated skeletal muscle regeneration, demonstrating that the inflammatory factors also support the activities of SCs [54,55]. Along this line, macrophages play a central role in regulating skeletal muscle regeneration by giving rise to M1 and M2 macrophages [56]. The M1 macrophages are pro-inflammatory, while the M2 macrophages are tissue-healing [57]. M1 macrophages are the predominant type of macrophages in the early phase of muscle regeneration. They remove muscle debris resulting from trauma and secrete cytokines, such as tumor necrosis factor α (TNF-α) and Interleukin 6 (IL-6), which amplify the inflammatory status [58,59]. TNF-α attracts SCs to the injured muscle site and promotes the proliferation of SCs by activating the transcription factor nuclear factor-kappa B (NF-κB) [60]. In addition, TNF-α activates the p38 signaling pathway and stimulates SCs to differentiate [61]. Blocking TNF-α action by anti-TNF-α or inhibiting p38 kinase activity leads to downregulating the expression of muscle differentiation markers such as MYOD, MYOGENIN, or MHC [61,62]. IL-6 stimulates the migration, proliferation, and differentiation of myoblasts [63]. A reduction in IL-6 expression impairs differentiation, whereas overexpression of IL-6 enhances muscle stem cell differentiation [63]. Besides macrophages, T cells are the major cell population recruited to the lesion in the second wave of immune cell infiltration. Recently, it has been demonstrated that IL-1α, IL-13, TNF-α, and IFN-γ secreted by T cells are sufficient to promote SCs’ expansion both in vivo and in vitro [64]. Differentiation of SCs and maturation of newly formed myofibres is supported by the switch to an anti-inflammatory microenvironment promoted by M2 macrophages [65]. M2 macrophages produce anti-inflammatory cytokines, including IL-4, IL-10, and IL-13, to suppress the local inflammatory response at the injury site [66]. The conversion of M1 to M2 macrophages is facilitated by cytokines, such as IL-10, secreted by Regulatory T cells (Tregs) [67,68]. There is now evidence that M2 macrophages promote SCs’ differentiation into myotubes and thus support the late stage of myogenesis and regeneration [69,70]. The absence of M2 macrophages induces a delay in muscle growth and blocks muscle differentiation and regeneration [71]. Thus, this transition in macrophage phenotype is an essential component of muscle regeneration in vivo after acute or chronic muscle injury [68].

The regeneration process is a tightly regulated process in which the functions of SCs are regulated by factors secreted by other cells, such as fibroblasts and immune cells, giving these populations the same importance as SCs.

## 3. Skeletal Muscle Regeneration in Pathological Conditions

### 3.1. Age-Related Sarcopenia

In recent decades, the scientific community has begun to use the term “frailty” to refer to a progressive loss of functions, resulting in an inability to respond to physical or psychosocial stress [72,73,74]. One of the critical mechanisms of frailty is age-related sarcopenia, characterized by low muscle mass, low physical function (e.g., gait speed), and low muscle strength (e.g., grip strength) due to no other specific causes [75]. Possible explanations for the decline in muscle quality include (i) infiltration of fat in the muscles or myosteatosis [76], (ii) altered deposition of collagen and other non-contractile tissue in the muscles [77], and (iii) progressive loss of individual muscle fibers [6], including a decrease in the proportion of fast-twitch type II muscle fibers, while the size of type I muscle fibers remain largely unaffected during aging [78]. Finally, sarcopenia is associated with a reduced number of SCs, especially in type II fibers (fast), which is considered a major cause of physical disability and loss of independence in the geriatric population of humans and mice [79,80,81]. Indeed, if the number of SCs decreases, fewer stem cells are responsive in the tissue to activate the regeneration process and repair muscle injuries after trauma. In particular, Pax7^+^ SCs decrease during aging as they cycle more frequently during homeostasis and spend less time in a quiescent state, decreasing their ability to self-renew [82,83,84,85]. In this context, basal macroautophagy is a key system for maintaining organelle and protein homeostasis [86]. This activity gradually decreases during aging, leading to the accumulation of toxic cellular debris and dysfunctions [87]. In geriatric mice, the decrease in quiescent SCs has been associated with increased expression of the master regulator of senescence p16, leading to impairment of the regenerative apparatus [85]. An increase in p16 has been associated with a decrease in AUF1, an RNA-binding protein/cytokine mRNA destabilizing protein, which regulates several aging-associated pathways [88]. Analysis of EDL myofibers shows a significant age effect on the number of SCs per myofiber, with an age-related shift towards myofibers with fewer SCs [83]. The number of SCs per myofiber has been described to be significantly higher in two younger groups of mice (i.e., 3–4 and 7–10 months), but the median value of SCs per myofiber declined nearly to that of senile mice by 1 year of age (i.e., in the 11–13 months group) [83]. To confirm this aspect, aged GFP-SCs were transplanted into adult mice, and the SCs pool showed a decrease of 60% compared to young cells, suggesting age-dependent cellular dysfunction [80,84]. On the other hand, transplanting young SCs into the muscle of progeroid mice prolongs lifespan and ameliorates the alterations observed in skeletal muscle [89]. However, the reduced number of SCs is not the only cause of the decline in skeletal muscle regeneration with age (Figure 1).

In addition to the cellular dysfunctions of aged SCs, the muscle environment also modulates the activity of SCs. To this purpose, hetero-parabiosis and hetero-transplantation analyses have shown that the decline in muscle regeneration during aging can be partially reversed if aged SCs are transplanted into a youthful external environment [90]. This finding suggests that the dysfunctions of progenitor cell activity may be modulated by systemic factors that change with age. In this line, aged SCs show significant changes in TGF-β signaling, resulting in an increase in fibrotic phenotype and a decrease in myogenic differentiation through various mechanisms, including suppression of MYOD transcription [91]. Myostatin, also known as growth and differentiation factor 8 (GDF8), belongs to the TGF-β superfamily and inhibits muscle growth/development. Notably, myostatin serum levels increase with age, and muscle mass is inversely correlated with myostatin serum levels. This suggests a link between myostatin and age-related sarcopenia and opens up the possibility of developing therapeutic strategies to stimulate muscle growth and prevent muscle wasting [92]. In contrast, the pathway of the insulin-like growth factor 1 (IGF-1), which plays a central role in muscle growth, differentiation, and regeneration, is lower in the serum of aged individuals with poorer muscle strength, walking speed, mobility tasks, and various physical performances [93]. Indeed, overexpression of the IGF-1 isoform in muscle led to muscle hypertrophy in adulthood and guaranteed the maintenance of muscle mass and functionality during aging and in animal models of neuromuscular diseases [94]. Finally, aging is generally associated with a chronic state of slightly elevated plasma levels of pro-inflammatory effectors [95,96]. This condition is often referred to as low-grade inflammation (LGI), characterized by pro-inflammatory cytokines (TNF-α, IL-6, and NF-κB overactivation), which impair the functions of SCs and the efficacy of the regeneration process [97]. The alterations in the secretion of systemic factors may help to develop preventive approaches to monitor the progression of sarcopenia and intervene promptly.

In addition to changes in the microenvironment, another critical element responsible for SCs’ fate is the structural component of the stem cell niche. Stearns-Reider KM et al. have shown that aging is associated with increased muscle stiffness due to a pathogenic extracellular matrix (ECM) architecture, resulting in altered myogenic progression of SCs and affecting their behavior [98]. In particular, the niches of SCs are dramatically altered by abnormal deposition of ECM, leading to changes in the dynamics of cell support, resulting in a loss of function of SCs and, consequently, a decline in the regenerative capacity of skeletal muscle tissue [99]. In this context, some ECM components, including laminin, biglycan, and testican-1, are expressed at higher levels in old muscles than in young muscles under homeostatic conditions [99]. Moreover, lower fibronectin levels were found in old tissues than in young ones after injury [99]. The different effects on ECM composition could be due to the localization of individual components in the niches of SCs. Interestingly, the disruptions in the ECM converge on a few significant axes of intercellular communication, including integrin signaling, confirming previous findings demonstrating dysfunctional Itgb1 signaling in aging SCs [100]. The results showed that several signaling pathways are altered in aged skeletal muscle, leading to increased ligands and cell surface receptors on SCs [101].

FAPs in skeletal muscle tissue produce most of the ECM proteins, which show age-related differences. Since the number of FAPs does not increase with age, several groups have speculated that the gene expression programs of FAPs change, resulting in altered abundance of specific ECM molecules [53,101]. In particular, it has recently been shown that the ECM molecule Wisp1 is significantly reduced in the aged ECM of skeletal muscle and that this decrease is due to changes in aged FAPs [53]. At the same time, other mechanisms that are not dependent on transcriptional changes, such as altered protein turnover due to reduced degradation or increased protein synthesis and secretion, could also lead to protein accumulation in the ECM of aged skeletal muscle. In addition to the role that FAPs play in altering the composition of the extracellular matrix, their differentiation into adipocytes is enhanced, leading to the accumulation of ectopic fat in the interstitial space of muscle tissue, which contributes to the deterioration of muscle function with age. In particular, FAPs have been shown to respond to a gastric inhibitory polypeptide (GIP), a 42-amino-acid hormone released in response to the ingestion of nutrients, including fat or glucose. The inhibition of this metabolism reduces the accumulation of the intramuscular adipose tissue and improves sarcopenia [102] (Figure 1).

Finally, the inability to regenerate in a sarcopenic phenotype has been linked to the deterioration of neuromuscular junctions (NMJs), leading to chronic denervation. Presynaptic nerve terminals from mice at 29 months of age were disorganized and exhibited extensive sprouting compared to mice at 3 months of age. Postsynaptic endplates appeared diffuse, irregular, and with granular fragmentation [103]. The persistent changes in NMJs in aged mice reflect an alteration of the SCs pool, leading to impaired regenerative machinery. In particular, uncontrolled proliferation of SCs occurs after denervation injury, which could be responsible firstly for the exhaustion and then for the depletion of the SCs pool [104]. Indeed, SCs are reduced by one-third after 7 months of denervation compared to uninjured mice [104]. The reduction of SCs in denervated muscle was directly related to injury due to an increased susceptibility to apoptotic cell death [105] (Figure 1).

Based on these data, it is clear that a tight balance of several players modulates the activity of SCs. If one of them does not correctly work, the whole machinery can be impaired.

In recent years, several clinical trials on the treatment of sarcopenia have shown that exercise is the only effective strategy to counteract sarcopenia [106]. In particular, resistance and eccentric training are recommended as first-line therapy for the treatment of sarcopenia, as they improve muscle mass, strength, and physical performance in older people [107,108,109,110]. The beneficial effects of resistance training on muscle regeneration are associated with an increase in the number of SCs in humans [111]. The same results have been shown in mouse and rat models [112,113]. Specifically, a 13-week moderate-intensity run in rats of both sexes leads to an increase in the number of SCs per myofiber [83]. Exercise increases the number of SCs and enhances their ability to differentiate into structural and functional myotubes in vitro [112]. Current understanding of the factors that activate SCs during exercise is limited. Some results show that SCs activation during exercise is mediated by elements released by myofibers and interstitial cell populations, including IGF-1 and molecules secreted by FAPs supporting SCs in the regeneration process [114,115]. Based on these findings, the World Health Organization and, in particular, the Global Recommendations on Physical Activity for Health, recommend that adults 65 years and older should perform 150 min of moderate or 75 min of vigorous-intensity aerobic activity and two or more days of muscle-strengthening activities per week [116].

### 3.2. Duchenne Muscular Dystrophy

Muscular dystrophies are a heterogeneous family of neuromuscular disorders characterized by muscle wasting. Different dominant and recessive genetic mutations lead to dystrophies with different prognoses and phenotypes [117]. In most diseases, a mutation is present in genes encoding proteins of the dystrophin-associated glycoprotein complex that stabilizes muscle fibers during contraction by acting as a mechanical link between the cytoskeleton and the extracellular matrix [117]. Duchenne muscular dystrophy (DMD) is the most severe and common form of muscular dystrophy, affecting 1 out of 3500 male newborns [118]. The primary cause of the disease is a mutation in the gene encoding for dystrophin, resulting in a dysfunctional or absent protein [119,120]. Dystrophin links actin to the dystrophin-associated protein complex, and its lack leads to tears in the sarcolemma membrane and myofibers damage [121].

In addition to the primary cause, DMD is associated with high levels of oxidative stress and inflammation, which actively exacerbate the dystrophic phenotype. Although oxidative stress and inflammation are crucial for tissue regeneration after initial damage, their constant presence in skeletal muscles leads to further necrosis, injury, and accumulation of fibrosis [122,123,124,125,126,127]. Sustained and excessive infiltration of macrophages, as described in dystrophic muscles, disrupts redox homeostasis and causes muscle damage [128].

Among the causes of muscle wasting in muscular dystrophy, the dysfunctionality of SCs plays a crucial role. Understanding if this dysfunctionality can be counted among the secondary causes or if the dysfunctionality of SCs is strictly dependent on inflammation and oxidative stress, resulting in tertiary causes of DMD, is still unclear.

After an injury, SCs should be activated and differentiated to replenish damaged myofibers. However, the accumulation of fibrotic and adipose tissue may indicate that SCs cannot adequately replace damaged myofibers. The contribution of SCs to the progression of DMD has been extensively investigated. SCs’ dysfunction has been described since the 1980s [12,129], and a widely accepted model is the SC exhaustion caused by repeated cycles of muscle degeneration and regeneration leading to depletion of the regenerative capacity of SCs [130]. However, recent studies have shown that the number of SCs is increased in DMD patients and dystrophin-deficient mdx mice, suggesting that the defective regenerative capacity of dystrophic muscles is not exclusively due to a lack of SCs [131,132,133]. Genome-wide metabolic modeling and functional analyses have demonstrated defects in myoblast commitment. Indeed, 170 genes showed altered expression in mdx myoblasts compared to controls, including MYOD, MYOGENIN, Myomaker, Myomixer, epigenetic regulators, extracellular matrix interactors, calcium signaling, and fibrosis genes [134]. Functionally, increased myoblast proliferation, altered chemotaxis, and accelerated differentiation have been described in myoblasts isolated from dystrophic mice and patients [134]. In vitro differentiation of human pluripotent stem cells lacking dystrophin and DMD myoblasts displayed mis-organization of sarcomeres, contributing to myofibers instability, mis-localization of proteins belonging to the dystrophin-associated complex, myotubes branching, contraction defects and hyperactivation of calcium signaling [135,136,137].

The changes in the phenotype of SCs have been associated with both intrinsic and extrinsic defects. In particular, inflammation and oxidative stress trigger senescence and alter the plasticity of muscle stem cells [138,139,140]. Senescence can also be triggered by several cytokines, including TNF-α and TGF-β, elevated in the skeletal muscle of DMD patients [141]. TGF-β is a key mediator of fibrogenesis and acts as an inhibitor of SCs’ proliferation, activation, and differentiation [142,143]. TNF-α has bimodal effects on myogenesis depending on the concentration. Lower doses (0.05 ng/mL) of TNF-α stimulate myogenesis, while under pathological conditions, TNF-α levels between 0.5 and 5 ng/mL inhibit myogenesis [61].

Moreover, telomere shortening is one of the causes of senescence, and ablation of telomerase in dystrophic mice resulted in a more severe phenotype caused by impaired proliferation and a reduction in the number of SCs [130]. The accumulation of senescent cells expressing CDKN2A, p16, and p19 has been described to contribute to fibrosis and inflammation [144]. Along this line, the depletion of senescent cells by senolytic drugs improves muscle regeneration in dystrophic mice by decreasing the expression of IL-6, TGF-β, and IL-1β [144,145]. Furthermore, preventing senescence increases the regenerative potential of SCs and improves muscle functions [146]. In addition to senescence, altered plasticity of SCs has also been demonstrated over the years. In response to increased TGF-β signaling, myogenic cells adopt fibrogenic properties, with a decrease in PAX7 and MYF5 expression and an increase in fibrogenic genes expression (i.e., collagen I, fibronectin, and α-smooth muscle actin), leading to being more prone to undergo osteogenic or adipogenic transformations and produce extracellular matrix proteins [147,148]. On the same line, the inhibition of the TGF-β signaling pathway improves the expression of PAX7 and MYOD in muscles [147]. Moreover, a recent publication found that a group of muscle stem cells that possess an immune signature (C1qa/b, Lyz2, and Cd53) is present in severe dystrophic muscles [149] (Figure 2). Functional validation must be performed to understand their role in dystrophy. Besides the functions of inflammation and oxidative stress triggering SCs dysfunctions, a recent article has shown that the defects in SCs phenotype are also related to myofiber stability. Indeed, in transplantation experiments, Hicks and collaborators have demonstrated that skeletal muscle progenitor cells (SMPCs) require the support of ACTC1+ fibers to sustain the functional engraftment of SMPCs and to support the stem cell state [150]. Therefore, secondary players of muscular dystrophy trigger senescence and altered plasticity of SCs. Still, the senescence itself can exacerbate the effects of inflammation and oxidative stress, rendering SCs senescence a secondary cause of DMD.

On the other hand, intrinsic defects of SCs are the leading cause of SCs’ dysfunctionality. Dystrophin is expressed in SCs and is associated with Par1b, an important regulator of cell polarity. Its absence in DMD results in a 5-fold reduction in the proportion of asymmetric division, leading to a lower number of myogenic progenitor cells available for efficient muscle regeneration [133]. Intrinsic defects decrease the generation of myogenic progenitor cells required for proper muscle regeneration after tissue damage. The rescue of asymmetric division in dystrophin-deficient satellite stem cells, with EGF treatment, enhances regeneration in skeletal muscle from mdx [151]. This limitation highly impacts muscle regeneration, leading to the identification of intrinsic defects as a secondary cause of DMD.

Besides the destabilization of muscle myofibers due to the destabilization of the dystrophin–glycoprotein complex, dystrophin and the related structural proteins are required for synaptic homeostasis [152]. Abnormal neuromuscular junction (NMJ) morphology has been described in dystrophic patients, and the muscle weakness of mouse models for DMD is associated with changes in NMJ and neuromuscular transmission failure [153,154,155,156,157,158]. However, changes in NMJ morphology result from cycles of degeneration and regeneration of the tissue rather than a secondary cause of muscular dystrophies [158]. Still, this hypothesis remains to be fully elucidated (Figure 2).

Although a consensus has emerged over the years to enhance the functions of SCs and stimulate regeneration in dystrophy to improve the dystrophic phenotype, recent evidence has shown that slowing down the degeneration–regeneration cycles may also help to preserve skeletal muscle in dystrophies [4,126,151,159,160,161,162,163,164]. Despite this debate, the scientific community agrees that skeletal muscle must be preserved by correcting the genetic defect of DMD. Any proposal to restore dystrophin expression by gene editing in vivo or stem cell transplantation requires a synergistic approach to preserve skeletal muscle in order to cure the disease [165,166,167,168].

### 3.3. Cachexia

Cachexia, from the Greek *kakos* “bad” and *hexis* “condition,” is an unintentional body weight loss of more than 5% in the past 6–12 months or the acquirement of a low body mass index that occurs in association with various chronic diseases, including cancer, diabetes, rheumatoid arthritis, kidney disease, heart failure, and chronic infections (such as HIV and tuberculosis) [14,169,170,171]. Cancer cachexia affects 50–85% of cancer patients, depending on the type and stage of cancer: a higher incidence has been described in patients with lung, gastrointestinal, and pancreatic cancers, and overall, most advanced cancer patients suffer from emaciation [172]. In parallel with tumor growth, most organs and muscles experience a decline and loss of functions. This mechanism is driven by systemic inflammation, which is one of the first central players in triggering cachexia [173].

Over the years, a long list of cytokines has been described to increase in the blood of cachectic patients and animal models. These include TNF-α, interleukins (IL-6; IL-1β and IL-1α), interferon-gamma (IFN-γ), leukemia inhibitory factor (LIF), and the TGF-β superfamily have been shown to mediate cancer-induced muscle wasting [174,175,176,177,178]. Cytokines and tumor factors affect different organs and tissues, leading to their wasting and modifying their functionality. In chronological order, after the affection of the immune system, the brain, pancreas, liver, adipose tissue, and gut are severely involved before the skeletal muscles [173]. However, skeletal muscle wasting and dysfunctionality highly impact the quality of life and survival of patients, rendering the wasting of skeletal muscle central in the diagnosis of cachexia.

Over the years, the importance of the imbalance between protein synthesis and degradation has been described as a primary cause for the establishment of cachectic muscles. In this context, the two major pathways controlling protein synthesis are the Akt-mTOR pathway, which acts as a positive regulator of protein synthesis, and the myostatin-Smad2/3 pathway, a negative regulator of the IGF-1/Akt/mTOR pathway [179]. Conversely, the main processes controlling protein degradation are the ubiquitin-proteasome (UPS) and the autophagy-lysosome systems. In cancer cachexia, the loss of skeletal muscle is due to both a downregulation of protein synthesis, mainly due to the decrease in IGF-1 and the increase in myostatin pathway, and an upregulation of protein degradation due to the overexpression of atrogenes and the hyperactivation of the autophagy and proteasome system [180]. Specifically, atrophy occurs in type II fibers with a 40% reduction in diameter, while type I fibers remain largely unchanged [181], similar to fasting [152]. In addition, recent evidence has shown that the bone morphogenetic protein (BMP) pathway in cancer cachexia disrupts neuromuscular junction and denervation, contributing to muscle wasting [182]. IL-6 and Activin A have a synergic role in inducing Noggin overexpression that causes NMJ impairment and atrophy, whereas treatment aimed to increase BMP signaling attenuate muscle atrophy during cancer cachexia [182] (Figure 3).

In addition to the contribution of the balance between protein synthesis and degradation, several articles have demonstrated a central role of impaired regeneration in the establishment of cachexia in skeletal muscle, pointing to new important mechanisms in cachexia. Under atrophic conditions, the muscle membrane is damaged, as evidenced by alteration of the sarcolemma and basal lamina, which can be observed by increased Evans blue dye penetration, decreased expression of extracellular matrix genes, and increased IgG and diffuse laminin staining [181,183]. After the establishment of muscle damage, the regeneration program begins, as evidenced by myofibers with centrally located nuclei, leading to the expression of embryonic MHC and the formation of small muscle fibers [104,184,185]. However, clear evidence of defect of regeneration was demonstrated after acute injury, where regeneration in cachectic muscle is highly impaired, as evidenced by patches of necrotic fibers surrounded by few inflammatory cells [183,186,187]. Moreover, it has been demonstrated that fibrosis and collagen deposition increase in cachectic muscles in patients and mice to replace myofibers, suggesting that the progressive development of fibrosis is a consequence of the expansion and differentiation of FAPs progenitors in cachectic muscles [188,189,190,191]. This is also accompanied by no inflammatory infiltration in the muscle of tumor-bearing mice [192,193]. The infiltration of immune cells is fundamental for the proper timing of regeneration, and its deficiency in cachectic muscles could be one of the causes of poor regeneration [54]. Accordingly, IL-4 administration improves regeneration by impacting SCs’ numbers and functions [194]. The impaired regeneration is due to different cell populations whose functionality is strongly modified by tumor growth (Figure 3).

In addition to the impairment of homeostasis of myofibers, immune cells, and FAPs, attention has also focused on the effects of cachexia on the expression of PAX7. Indeed, results have shown that PAX7 expression is increased 10-fold in cachectic muscles compared to control subjects [183]. Independent of the atrophic injury, an increase in PAX7^+^ cells is associated with a higher number of SCs [183,195]. Moreover, interstitial localized cells, such as FAPs and pericytes, begin to express PAX7 and contribute to its overexpression [183]. The overexpression of PAX7 and an increase in the number of SCs without an increase in MYOD and MYOGENIN demonstrate the inability of SCs to differentiate into muscle fibers and repair the damage caused by atrophy while promoting self-renewal [183,196,197]. In addition, the accumulation of SCs is central to establishing the atrophic phenotype of muscles, and it is considered to be one of the main leading causes of cachexia [187]. Indeed, the administration of MEK inhibitors attenuates atrophy by restoring the myogenic potential of myoblasts [187].

Next, the question of the mechanism of the SCs’ dysfunction arises. Indeed, the dysfunction of SCs is closely related to the tumor because SCs readily fuse into muscle fibers after resection of the tumor [183]. Moreover, isolated SCs from tumor-bearing mice cultured in a differentiation medium without prior expansion showed a higher fusion index than cells isolated from control mice, suggesting that myogenic cells exposed to cachectic factors retain the ability to differentiate [183]. In vitro studies suggest that tumor factors can induce apoptosis and thus modulate the pool of functional SCs [198,199]. The mechanism by which SCs experience impaired differentiation capacity is due to the overexpression of PAX7 through the activation of the canonical NF-κB pathway [183]. Indeed, overexpression of PAX7 in vivo results in decreased muscle mass and increased muscle wasting in cancer cachexia [183]. Moreover, cachectic muscles depleted of PAX7 retain more muscle mass than control subjects [183]. PAX7 overexpression inhibits differentiation by suppressing MYOD and MYOGENIN, limiting myoblast fusion with damaged myofibers to complete regeneration [196]. Activation of NF-κB in myoblasts has been shown to inhibit MYOD synthesis and upregulate PAX7 [200,201]. In addition, inhibition of NF-κB signaling rescues cancer-related atrophy associated with increased fusion of SCs into muscle fibers [183]. At the same time, selective activation of NF-κB exacerbates muscle mass decline and increases PAX7 expression [183]. However, the mechanism by which NF-κB signaling is activated has not been fully elucidated, as serum from cachectic mice and patients induce PAX7 expression, whereas cytokines (TNF-α, IL-1β, IL-6, and IFN-γ, administered either singly or as a cocktail) or C-26 or LLC conditioned medium do not recapitulate the effect on PAX7 and further analyses are required to address this point. Another signaling pathway that negatively regulates the differentiation of SCs is the myostatin/ActRIIB pathway [183,202]. Blockade of the ActRIIB pathway reverses muscle wasting in tumor-bearing mice by attenuating protein catabolism and stimulating the growth of SCs [202]. In addition, the state of quiescence or hyper-quiescence of SCs could be beneficial to circumvent death. Increased expression of CCAAT/enhancer binding protein beta (C/EBPβ), an inhibitor of myogenesis, was found in SCs from cachectic animals. Genetic depletion of C/EBPβ leads to increased apoptosis and further impairment of regeneration. Taken together, the inhibition of muscle regeneration during cancer cachexia could be the result of tumor factors stimulating muscle wasting and degeneration, but also a mechanism of muscle to maintain viability and avoid death (Figure 3).

Recently, exercise, especially a combination of endurance and resistance training, has been shown to counteract or limit skeletal muscle atrophy in cachectic conditions [203,204,205,206]. The beneficial effects of moderate exercise have been explained by reduced inflammation, oxidative stress, and improved antioxidant capacity [204,207,208]. Moreover, physical training promotes hypertrophy mainly through the activation, proliferation, and differentiation of SCs [209]. In summary, various processes occur in skeletal muscle during cancer cachexia. In addition to modulation of myofibers size, deregulation of SCs functions has been described recently and can be exploited therapeutically. In particular, strategies aimed at restoring regeneration may also prove effective in restoring muscle mass in cancer patients.

## 4. Summary and Future Prospective

Muscle regeneration is a multi-faceted and well-orchestrated phenomenon in adulthood that regulates skeletal muscle tissue homeostasis and ensures the restoration of mass and function. Under physiological conditions, SCs play an essential role in providing the progeny required for regenerating all damaged tissue by switching from a quiescent to an active state. Each phase of stepwise muscle healing, including the expansion of SCs (PAX7^+^/MYOD^−^), differentiation into myoblasts (PAX7^−^/MYOD^+^), and the formation of new fibers, is characterized by the combination of activation and recruitment of cells and molecular signals playing specific roles in the complex framework of regeneration [20]. It is well known that changes in the number and function of SCs affect skeletal muscle regeneration in various pathological conditions. Based on these premises, the identification of new therapeutic approaches that can counteract the loss of SCs and reestablish their function is a strategy for treating various muscle diseases. In particular, a reduced number of SCs has been associated with the aging process leading to sarcopenia [11,83]. Indeed, SCs are lower in older people than in young controls in this condition. However, the effects of sarcopenia are related to the number of SCs and to a change in their function, leading to the wasting of skeletal muscle tissue [91]. The impairment of SCs’ function also plays a crucial role in muscular dystrophy, where the number of SCs seems to increase without compensating for the loss of muscle fibers replaced by an enhancement of adipose and fibrotic tissue deposition [131,132,133]. Along this line, cachectic patients also show an increase in SCs number without an increase in MYOD and MYOGENIN, demonstrating the inability of SCs to differentiate into muscle fibers, leading to skeletal muscle atrophy [183,195].

The alteration in the functions of SCs is mainly due to an increased inflammatory state in skeletal muscle tissue. In particular, the low chronic inflammation during aging induces SCs to move toward a senescence phenotype that increases the secretion of inflammatory cytokines such as TNF-α and IL-6 [95,96]. On the other hand, skeletal muscle dystrophy is characterized by an increased number of inflammatory cells, while cancer cachexia does not affect the cellular component but induces a significant enhancement of secreted cytokines, aggravating the pathological phenotype by increasing autoinflammation and fibrotic tissue deposition [123,124,175,176]. Another factor secreted by inflammatory cells that plays a central role in altering muscle regeneration is TGF-β. In this context, TGF-β promotes fibrogenesis by inhibiting the proliferation, activation, and differentiation of SCs and inducing FAPs cells to adopt a fibrogenic phenotype that promotes the deposition of ECM proteins [91,147,148].

In recent years, skeletal muscle tissue has been described as an autocrine, paracrine, and endocrine organ that regulates the functions of other organs [210,211]. Physical activity, in particular, stimulates this special feature through the release of proteins known as myokines. According to current knowledge, the primary physiological role of myokines is to protect the functionality of skeletal muscles [210,211]. Myokines control adaptive processes in skeletal muscle by acting as paracrine regulators of fuel oxidation, hypertrophy, angiogenesis, inflammatory processes, and regulation of the extracellular matrix [211]. The endocrine functions attributed to myokines regulate body weight, low-grade inflammation, insulin sensitivity, tumor growth suppression, and cognitive function improvement [211]. For these reasons, it is clear that physical activity can counteract muscle wasting in various skeletal muscle diseases. In addition to the classic approaches to reduce muscle atrophy, such as nutritional integration or pharmacological treatment, the development of therapeutic methods based on the identification of individualized exercise programs has become more attractive in the treatment of any pathological condition. In particular, several studies have investigated the effects of different types of exercise on the activation of SCs. In particular, eccentric exercise has been shown to stimulate the activation and proliferation of SCs in the elderly [110]. Along this line, acute activation of SCs after exercise leads to a regenerative response associated with a subsequent return to a quiescent state. This notion is supported by studies showing benefits to overall muscle health in the elderly who perform resistance training that stimulates SCs activation through the release of myokines from myofibers and interstitial cell populations (FAPs), which reduces chronic low-level inflammation and promotes SCs functions [212]. The role of exercise in reducing inflammatory levels and preserving SCs’ function is also critical in other pathological conditions. Indeed, aerobic exercise in muscular dystrophy and moderate training in cachexia counteract the inflammatory state, activating SCs and promoting skeletal muscle hypertrophy [204,207,208,209,213,214].

Currently, the understanding of SCs’ behaviors after exercise is relatively limited. Future studies need to precisely define the mechanisms involved in the changes of skeletal muscle regeneration under physiological and pathological conditions in order to design more appropriate training to counteract skeletal muscle atrophy and any systemic side effects on the organism.

## Figures and Tables

**Figure 1 ijms-25-00512-f001:**
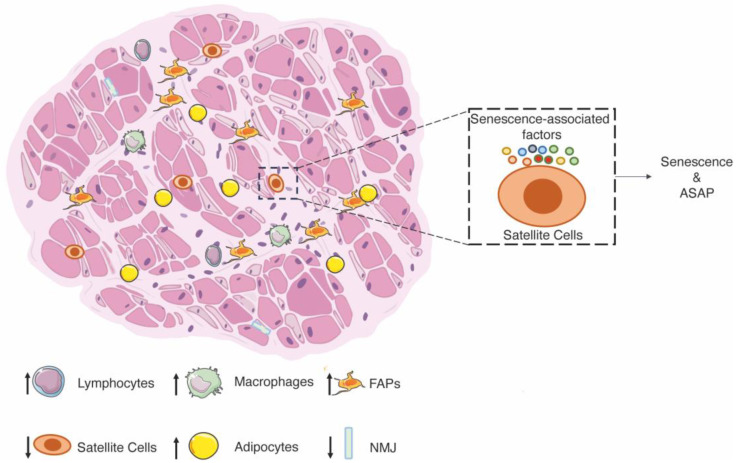
Schematic representation of sarcopenia. Sarcopenic skeletal muscle is characterized by the loss of myofibers, which are replaced by fibrous and fatty tissue. The number of SCs decreases in sarcopenia. The tissue is more heavily infiltrated with inflammatory cells (lymphocytes and macrophages). Fast fibers (light pink) are severely impacted by mass loss, while slow fibers are not affected (dark pink). The neuromuscular connections are impaired. On the right side, the SCs’ dysfunctions are caused by senescence-associated factors (higher concentration of cytokines and myostatin; decrease in IGF1), which stimulate senescence and the senescence-associated secretory phenotype.

**Figure 2 ijms-25-00512-f002:**
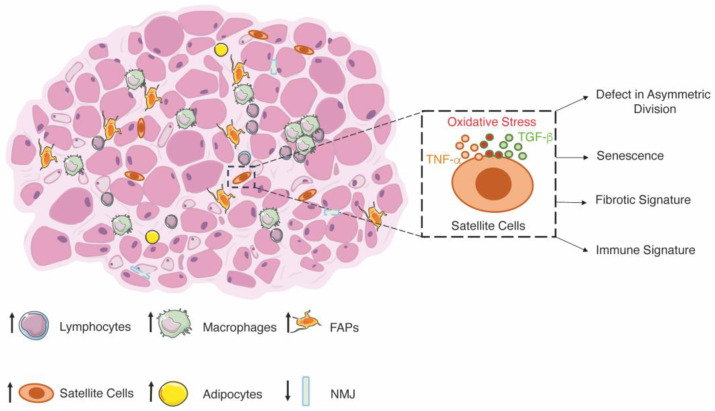
Schematic representation of muscular dystrophy. Dystrophic muscles are characterized by cycles of degeneration and regeneration, which, over time, lead to the loss of myofibers with an accumulation of mostly fibrotic tissue. The fast fibers (light pink) are the most affected by the cycles of regeneration and regeneration, causing an abnormal morphology of the neuromuscular junctions, while slow fibers are not affected (dark pink). High infiltration of inflammatory cells (lymphocytes and macrophages) leads to an increase in the number of SCs, which, however, show a dysfunctional phenotype. The dysfunctions observed in the SCs are triggered by extracellular factors (cytokines) but also by intracellular defects (lack of dystrophin) that lead to a disruption of asymmetric division as well as senescence and changes in the myogenic signature towards a fibrotic and immunological signature.

**Figure 3 ijms-25-00512-f003:**
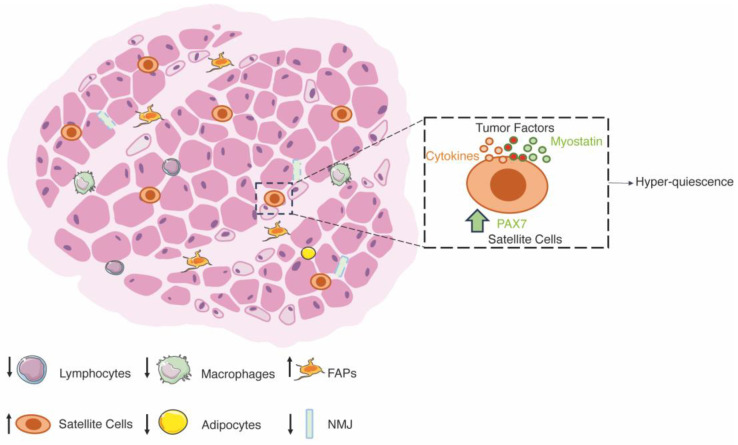
Schematic representation of cancer cachexia. A cachectic muscle is characterized by a decrease in muscle mass due to changes in metabolism but also by changes in the function of the SCs. The skeletal muscle shows a reduced number of infiltrating cells (lymphocytes and macrophages), although it is degenerated in places. The fast fibers (light pink) are most severely affected and show a high degree of denervation, while slow fibers are not affected (dark pink). The main change is a state of hyper-quiescence of the SCs, which leads to overexpression of PAX7 via the NF-κB and myostatin pathways.

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
