# Peer review of "Regulation of Satellite Cells Functions during Skeletal Muscle Regeneration: A Critical Step in Physiological and Pathological Conditions"

_ijms, 2023, doi:10.3390/ijms25010512_

Round 1

Reviewer 1 Report

Comments and Suggestions for Authors

Dear Editor-in-Chief

Thanks for your invitation to review the following review titled "Skeletal Muscle Regeneration: a Critical Step in Physiological and Pathological Conditions" . The review is well organized and has novel data. Unless the number of figures is low, please add a schematic figure of each section in the review.  Satellite cells should be included in the title. Please include the other pathological conditions as congenital,.......

Author Response

Point-to point Reviewer 1:

The authors appreciated the recognition of the importance of our results and are grateful for the suggestions that helped us to improve our work.

Major comments:

#1: Unless the number of figures is low, please add a schematic figure of each section in the review.

The authors worked on the figures and we included a schematic figure for each section in the revised manuscript, as suggested.  

#2: Satellite cells should be included in the title.

The authors modified the title of the review including the name “Satellite cells” as suggested.

#3: Please the other pathological conditions as congenital

The reviewer's comment is very interesting as congenital myopathies are a group of heterogeneous disorders characterized by hypotonia and weakness from birth with a static or slowly progressive clinical course (Cassandrini, D., et al. 2017). Recently, some of the myopathies have been classified as primary satellite cell-opathies, defined as diseases primarily caused by impaired satellite cell functions (Ganassi M et al 2022). EMARDD disease (Myopathy, Aareflexia, Respiratory distress, And Dysphagia, Early-Onset), for example, is caused by a mutation in MEFG10, a transmembrane protein involved in the proliferation and migration of satellite cells (Holterman C.E., et al 2007; Li C. et al 2021). Severe diaphragmatic dysfunction, weakness, and hypotonia characterize EMARDD patients who do not achieve independent walking (Harris E. et al 2018). MEFG10 remains to be fully explored to understand its role in developing this condition. Including congenital disorders, where the regenerative apparatus is less involved and less addressed in literature, might mean an increased risk of overwhelm the manuscript with a huge amount of scientific information without the opportunity to go into detail in each pathology. For this reason, the authors have chosen to address this manuscript by discussing only sarcopenia, cancer cachexia, and muscular dystrophy and the changes in the regeneration process. They focus on the role of satellite cells and address intrinsic and extrinsic changes that may negatively affect their ability.

Reviewer 2 Report

Comments and Suggestions for Authors

The review focuses mainly on the effect of different conditions and muscle mass rather than muscle regeneration. There is a part of the review on muscle regeneration; however, most of the study is devoted to sarcopenia and muscle mass loss in cancer cachexia. The mechanisms involved were partially described. Muscle mass and branch-chain amino acid metabolism were not considered, although it may be important. There are also important mechanisms which were not discussed i.e doi: 10.1016/j.steroids.2023.109328.

The genetics of muscle regeneration were partially discussed; important issues as those discussed recently doi: 10.1038/s41556-023-01271-0, doi: 10.1038/s41467-023-42837-8, 10.1038/s41419-023-06228-7 should be discussed therally. In addition, adipose tissue infiltration in sarcopenia, muscle waste in hypertension doi: 10.4415/ANN_23_03_10 and other muscle pathologies doi: 10.1007/s11914-022-00751-w and 10.1016/j.arcmed.2023.102890 should be discussed in detail since muscle contraction is affected. The authors did not include them ie doi: 10.1002/jcsm.13346 and the role of hnRNP  doi: 10.3390/biom13101434

I would suggest the authors modify the table and include the relevant issues raised. In addition, properly discuss the relationship between inflammation, adipocyte and leukocyte migration in different pathologies. 

Comments on the Quality of English Language

Minor grammatical mistakes were observed.

Author Response

Point-to point Reviewer 2:

The authors appreciated the recognition of the importance of our results and are grateful for the suggestions that helped us to improve our work.

Major comments:

#1: There are also important mechanisms which were not discussed (doi: 10.1016/j.steroids.2023.109328).

The authors added citation, as suggested. Due to paper revision, the new lines are 548-249

#2: The genetics of muscle regeneration were partially discussed; important issues as those discussed recently doi: 10.1038/s41556-023-01271-0, doi: 10.1038/s41467-023-42837-8, 10.1038/s41419-023-06228-7 should be discussed therally.

As suggested, the authors added the following citations.:

doi: 10.1038/s41556-023-01271-0 lines 462-469

doi: 10.1038/s41467-023-42837-8 lines 102-106

doi: 10.1038/s41419-023-06228-7 lines 188-191

#3: Adipose tissue infiltration in sarcopenia, muscle waste in hypertension doi: 10.4415/ANN_23_03_10 and other muscle pathologies doi: 10.1007/s11914-022-00751-w and 10.1016/j.arcmed.2023.102890 should be discussed in detail since muscle contraction is affected.

The authors thank the reviewer for the suggestion, but the manuscript focuses on muscle regeneration and changes in sarcopenia, cancer cachexia, and DMD. Sarcopenia is defined as primary sarcopenia when it is not associated with other pathologic conditions and secondary sarcopenia when it is due to the presence of a specific disease, such as obesity (Santilli V. et al 2014). In this review, the authors focused their interest on primary sarcopenia related to age. To avoid increasing the complexity of this manuscript, the authors have decided not to include the proposed article. In addition, the authors change the title of sarcopenia to "Age-related sarcopenia" (line 226).

#4: The authors did not include doi: 10.1002/jcsm.13346 and the role of hnRNP  doi: 10.3390/biom13101434

The authors added citation, as suggested. Due to paper revision, the new lines are 327-331. Moreover, the authors also included in the manuscript (lines 249-251) the role of hnRNP adding the reference 10.1016/j.molcel.2012.04.019, which specifically described the functions of AUF1 during sarcopenia.

#5: I would suggest the authors modify the table and include the relevant issues raised.

The authors decided to remove the table from the manuscript since it was only a repetition of reference already cited in the text, as suggested by another referee.

#6: Properly discuss the relationship between inflammation, adipocyte and leukocyte migration in different pathologies.

As suggested by the referee, the authors increased the discussion regarding fibrosis, adipocytes and inflammation adding details in the section related to sarcopenia (1676-1683), muscular dystrophy (line 1933-1949), and cancer cachexia (lines 2126-2140).

Reviewer 3 Report

Comments and Suggestions for Authors

In the current review, the authors summarized processes regulating satellite cell activation and skeletal muscle regeneration.

This is an interesting review, nevertheless, it required significant improvements. This review provides a listing of the previously published observations but lacks critical review and interpretation/ interconnection of these observations that characterize the well-written review papers. The authors should re-write the review to make it shorter, more focused, and better organized with interpretation /discussion of the most recent observations. Currently, it has a lot of space taken by the descriptions of well-known facts and general information that has no relevance to the topic of this review.

The language of the manuscript required significant improvement. For example: Line 34: “thus revealing its highly plasticity”.

Line 38: “when protein degradation signaling pathway is activated skeletal muscle tissue became atrophic”.

Line 455: “It Is evident that SCs are actively Involved In the phenotype of muscle dystrophies”.

Questions/ suggestions/ limitations of the study.

Abstract:

Line 12: Regular physical activity without significant muscle damage usually does not result in the generation of new myofibers.

Role of Satellite Cells in Regulating Skeletal Muscle Regeneration

Line 88: “Remarkably, quiescent SCs express the paired-box transcription factor PAX7 but also PAX3, which is particularly abundant in the diaphragm but much less abundant in hindlimb muscles”. Please provide your opinion on why this is observed.

Line 162: “This process is followed by maturation into contractile myofibers and in the expression of adult MHC isoforms with temporally and spatially regulated expression patterns.” Please describe the role of innervation in this process.

Figure 1 does not provide any new/important information. It was published in one form or another in many previous reviews.

Lines 230-240: Is this paragraph necessary? The focus of the review is on skeletal muscle regeneration. Similar examples of unnecessary information can be found in other places in this review.

One of the major causes of sarcopenia is progressive muscle fiber denervation. There is no mention of these changes in this review.

Table 1 is not informative. It repeats what already was described in the text, including the references.

Figure 2 should reflect fiber-type changes and denervation status.

Comments on the Quality of English Language

The language of the manuscript required significant improvement.

Author Response

Point-to point Reviewer 3:

The authors appreciated the recognition of the importance of our results and are grateful for the suggestions that helped us to improve our work.

Major comments:

#1: The language of the manuscript required significant improvement.

As suggested by the referee, the authors revised all the manuscript improving the language and adding the specific requested listed in the comment:

  • Line 34: "thus revealing its highly plasticity" was modified in "Skeletal muscle requires large amounts of energy and proteins for its functions but is also a source of energy during starvation and disease and, therefore, shows high plasticity"
  • Line 38: "when protein degradation signaling pathway is activated skeletal muscle tissue became atrophic" was modified in "When protein synthesis exceeds protein degradation, skeletal muscle grows and becomes hypertrophic. In contrast, hyperactivation of the protein degradation signaling pathway leads to an atrophic phenotype of skeletal muscle tissue."
  • Line 455: "It Is evident that SCs are actively Involved In the phenotype of muscle dystrophies" was removed by the manuscript

#2: Line 12: Regular physical activity without significant muscle damage usually does not result in the generation of new myofibers.

As suggested by the referee is not properly correct affirm that physical activity induces muscle damage. For this reason, the authors added to the abstract the term "competitive".

#3: Line 88: "Remarkably, quiescent SCs express the paired-box transcription factor PAX7 but also PAX3, which is particularly abundant in the diaphragm but much less abundant in hindlimb muscles". Please provide your opinion on why this is observed.

The authors provided their opinion regarding the reason why PAX3 in abundant in diaphgram but less in the hindlimb muscles (lines 152-154).

#4 Line 162: "This process is followed by maturation into contractile myofibers and in the expression of adult MHC isoforms with temporally and spatially regulated expression patterns." Please describe the role of innervation in this process.

The authors thank the referee for the suggestion. The authors added the role on innervation in expression of MHC isoform (line 864-869).

#5: Figure 1 does not provide any new/important information. It was published in one form or another in many previous reviews.

The authors removed this figure and added one figure in each section of the manuscript.

 #6: Lines 230-240: Is this paragraph necessary? The focus of the review is on skeletal muscle regeneration. Similar examples of unnecessary information can be found in other places in this review.

The authors thank the referee for the suggestion and decided to remove paragraph indicated by the referee and also others in each section of the manuscript, as can be observed in the revised version.

#7: One of the major causes of sarcopenia is progressive muscle fiber denervation. There is no mention of these changes in this review.

The authors thank the referee for the suggestion. The authors added the role on innervation during sarcopenia (line 1688-1702).

#8: Table 1 is not informative. It repeats what already was described in the text, including the references.

The authors decided to remove table 1 from the manuscript.

#9: Figure 2 should reflect fiber-type changes and denervation status.

The authors removed this figure and added one figure in each section of the manuscript, in which are also present the changes in the innervation status in each pathological condition.

Round 2

Reviewer 2 Report

Comments and Suggestions for Authors

Now the article is much better. There are some grammatical mistakes in the text. The authors some modifications, and the manuscript was improved partially. There are still some editorial changes to be made

Comments on the Quality of English Language

Several sentences require attention.

Reviewer 3 Report

Comments and Suggestions for Authors

My previous recommendation still stands: the authors should re-write the review to make it shorter, more focused, and better organized with interpretation/discussion of the most recent observations. It looks like the authors made the review even longer adding more information. A good review is not characterized by its length and whether or not it lists all of the previously published facts. It is the ability of the authors to interpret the large amount of previously published data and present it in a clear and concise way pointing out the most important recent observations.  This manuscript does not have these qualities. The current review lists a large number of small details of diverse studies but fails to present a cohesive and clear story of satellite cell regulation.

The revised figures are not informative and do not help the understanding of the main message that the authors tried to convey in the particular chapter.

The language of the revised manuscript requires significant improvement. Please see some of the examples, revisions are required throughout the entire manuscript, and too many changes are needed to list all of them.

Line 20: “Understanding the molecular mechanisms involved in their dysregulations is relevant to improve current treatments”.

Line 39: “the well-being of skeletal muscle is closely linked to its ability to repair and regenerate new muscle fibers after blunt or penetrating traumatic events such as road traffic”.

Comments on the Quality of English Language

The language of the revised manuscript requires significant improvement. 

Round 3

Reviewer 2 Report

Comments and Suggestions for Authors

The article was improved. It should be published as is.

Comments on the Quality of English Language

Minor grammatical mistakes